# Low ocean-floor rises regulate subpolar sea surface temperature by forming baroclinic jets

H. Mitsudera [1], T. Miyama [2], H. Nishigaki [3], T. Nakanowatari[4], H. Nishikawa[1], T. Nakamura[1], T. Wagawa[5], R. Furue[2], Y. Fujii[6] & S. Ito[7]

Sea surface temperature (SST) fronts in mid- to high-latitude oceans have significant impacts on extratropical atmospheric circulations and climate. In the western subarctic Pacific, sharp SST fronts form between the cold subarctic water and the recently found quasi-stationary jets that advect warm waters originating in the Kuroshio northeastward. Here we present a new mechanism of the jet formation paying attention to the propagation of baroclinic Rossby waves that is deflected by eddy-driven barotropic flows over bottom rises, although their height is low (~500 m) compared with the depth of the North Pacific Ocean (~6000 m). Steered by the barotropic flows, Rossby waves bring a thicker upper layer from the subtropical gyre and a thinner upper layer from the subarctic gyre, thereby creating a thickness jump, hence a surface jet, where they converge. This study reveals an overlooked role of low-rise bottom topography in regulating SST anomalies in subpolar oceans.

[1] Pan Okhotsk Research Center, Institute of Low Temperature Science, Hokkaido University, Sapporo 060-0819, Japan. [2] Application Laboratory, Japan Agency for Marine-Earth Science and Technology, Yokohama 236-0001, Japan. [3] Division of Natural Sciences, Faculty of Science and Technology, Oita Univeristy, Oita 870-1192, Japan. [4] National Institute of Polar Research, Tachikawa 190-8518, Japan. [5] Japan Sea Fisheries Research Institute, Japan Fisheries Research and Education Agency, Niigata 951-8121, Japan. [6] Meteorological Research Institute, Japan Meteorological Agency, Tsukuba 305-0052, Japan. [7] Atmosphere and Ocean Research Institute, The University of Tokyo, Kashiwa 277-8564, Japan. Correspondence and requests for materials should be addressed to H.M. (email: humiom@lowtem.hokudai.ac.jp)

Recent high-resolution satellite observations and numerical models have revealed that sea surface temperature (SST) variations associated with eddies and frontal jets in the mid- to high-latitude oceans can cause significant far-reaching influences on extra-tropical atmospheric circulations[1–7]. In the western North Pacific, SST variations are pronounced between 40°N and 43°N where SST gradients are large[4,8]. This region is often referred to as the Subarctic Frontal Zone (SAFZ), which corresponds to the boundary between the subtropical and sub-arctic gyres[9,10]. The SST anomalies in the SAFZ likely induce anomalous atmospheric circulations in the Northern Hemisphere spanning from the western Pacific to the North America and even to the western Europe, and can cause significant impacts on climate variability in distant regions such as winter rainfall along the Pacific coast of the North America[7]. Numerical experiments using high-resolution atmospheric models indicate that sharpness of the SST fronts in the SAFZ is essential for generating these atmospheric responses[5,6]. However, our understanding to a fundamental question, as to how such non-zonal sharp SST fronts can form and stay almost stationary away from the western boundaries of oceans, is still lacking.

To solve the above question and elucidate the front formation, we pay attention to recently found quasi-stationary jets in the SAFZ[11,12], which flow northeastward and transport the warm water originating in the Kuroshio Extension. Isoguchi et al. identified two jets[11], one of which is located around 150°E–155°E and the other is located around 165°E–170°E in the SAFZ. These jets are exhibited robustly in ocean current products such as that in Fig. 1a from Japan Coastal Ocean Prediction Experiment 2[13] (JCOPE2). For clarity, we refer to these jets as the Isoguchi J1 and J2 according to their definition. Sharp SST front forms where the warm J1 and the cold Subarctic Current (that is regarded as an Oyashio's extension) are confluent; the SST gradient there exceeds 5 °C per 100 km across the front[8]. The Isoguchi J1 and the SST front appear to form associated with bottom topography[9,11,14] (Fig. 1b). However, it is puzzling why such a small-amplitude topography (~500 m) can anchor the jet in the deep North Pacific Ocean (~6000 m).

Here we present a mechanism of the northeastward Isoguchi jet formation from a point of view of baroclinic Rossby wave propagation, the pathways of which, which are characteristic curves, are deflected by ambient barotropic flows. Because of the deflection of characteristic curves, a thin upper layer originating in the subarctic gyre and a thick layer originating in the subtropical gyre converge at locations where Isoguchi jets exist, thereby creating thickness jumps and causing the baroclinic jets. The ambient barotropic flows are generated by eddy-topography interaction over topographic features of ~500 m, which consequently regulate the SST fronts in the SAFZ in the North Pacific Ocean. This study reveals an overlooked role of small-amplitude bottom topography in the formation of surface jets and SST fronts in subpolar oceans.

## Results

**Subarctic SST fronts and quasi-stationary Isoguchi jets**. We first examine the relationship between the SST front variability and the Isoguchi Jets. The SST front variability is evaluated by the empirical orthogonal function (EOF) of the latitude $\Phi(x, t)$ representing the maximum meridional SST gradient ($d\mathrm{SST}/dy$) in the region 145°E −170°E, 35°N−47°N, where $x$, $y$, and $t$ denote zonal coordinate, meridional coordinate, and time, respectively. The time coefficient of the 1st EOF mode is known as the Oyashio Extension Index (OEI)[4,15] which exhibits significant correlation with an atmospheric teleconnection, the North Pacific Oscillation/Western Pacific Pattern[16], with a lead time of a few months[4]. The temporal mean SST front position $\overline{\Phi(x)}$ (thin white line in Fig. 1a) derived from JCOPE2, as well as from the Optimum Interpolated SST (OISST)[17] (thin black line in Fig. 2a), coincide with the J1 in the longitude range 150°E−155°E and the J2 in 165°E−170°E. Further, north–south shift of the SST front, represented by the 1st EOF mode (denoted by difference between thin line and thick line), is small in these longitude ranges where the J1 and J2 flow, consistent with stationary nature of these jets[8]. On the other hand, the SST front between 155°E and 165°E exhibits a large north–south shift. We note that the J1 crosses $\overline{\Phi(x)}$ around 42°N, 155°E where $\overline{\Phi(x)}$ reaches the northernmost position. This suggests that strong J1 would produce warm SST anomaly downstream by moving the SST front, or the maximum $d|\mathrm{SST}|/dy$ position, northeastward.

We thus examine the jet-SST relationship further by evaluating the J1 strength with the satellite SSH difference between the two

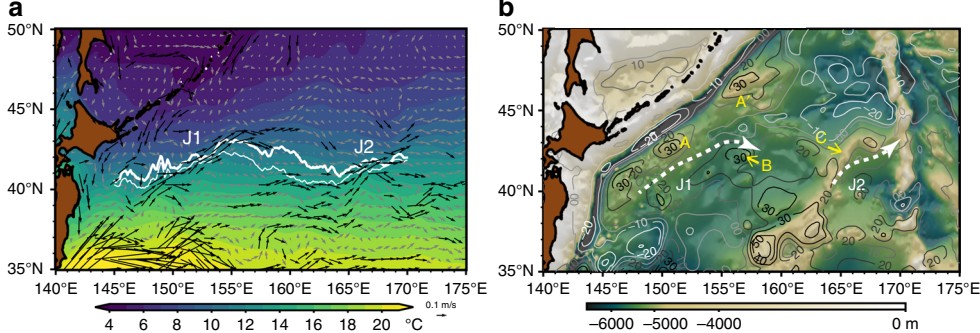

**Fig. 1** Quasi-stationary jets in the Subarctic Frontal Zone. **a** Annual mean SST (shade), surface current (arrows) and SST front position (white solid lines) derived from the JCOPE2 ocean reanalysis (see Methods). Surface currents faster than 0.1 ms$^{-1}$ are denoted by black arrows, while slower than 0.1 ms$^{-1}$ by gray arrows. As for the SST front position, it is represented by the latitude $\Phi(x, t)$ where the meridional gradient of SST is maximum in the region 145° E−170°E, 35°N−47°N, derived from JCOPE2. Here, $\Phi(x, t)$ is decomposed to $\Phi(x,t) \sim \overline{\Phi(x)} + \mathrm{EOF1}(x) \cdot \mathrm{OEI}(t)$, where $\overline{\Phi(x)}$ is the temporal mean position, and EOF1($x$) and OEI($t$) denote the 1st mode empirical orthogonal function (EOF) and its time coefficient, respectively. OEI($t$) is normalized so that its standard deviation $\sigma$ to be unity. White solid lines denote $\overline{\Phi(x)} + \mathrm{EOF1}(x) \cdot \sigma$, where $\sigma = 0$ (thin line) and $\sigma = 1$ (thick line). **b** Barotropic streamfunction (contours, ×10$^6$ m$^3$ s$^{-1}$) and bottom topography (shades). Barotropic streamfunction is obtained from depth-integrated transport using horizontal velocity field of JCOPE2 reanalysis. Black (light blue) contours denote the positive (negative) value in terms of the streamfunction. The bottom topography is based on ETOPO5, in which bathymetry between 5500 m and 5000 m is emphasized. **A**, **A'**, **B**, and **C** denote the ocean-floor rises that are paid attention in this paper. J1 and J2 are indicated by dashed lines with arrows

boxes located across J1 (Fig. 2a). The SSH difference is referred to as the Isoguchi Jet Index (IJI; Fig. 2b), which exhibits pronounced decadal-scale variations (>7 year period) that contain 55% of total energy. The IJI is correlated significantly with the OEI (Supplementary Figure 1), where correlation coefficient is 0.64 in winter (December, January, February) with the significance level exceeding 95%. The regression map between the wintertime IJI and the SST anomaly (Fig. 2a) resembles the SST regressed to the OEI[6,18]. The IJI–SST relationship indicates that SST eastward of J1 increases as the IJI increases (Fig. 2a), consistent with the anomalous SST formation by anomalous heat advection. Significant correlation with large regression coefficients is seen around 42°N, 153°E over the SST front. Other significantly correlated SSHs with large regression coefficients are found around 42.5°N, 157°E and 43°N, 160°E north of the mean SST-front latitude $\overline{\Phi}(x)$, suggesting the cross frontal heat transport by the J1 as suggested in the preceding paragraph.

Anomalous surface flows that cause heat advection may be seen from the satellite-derived SSH anomaly. Figure 2c indicates that the SSH anomaly of decadal time scale (>7 years), correlating positively with the IJI, extends northeastward from the south-eastern box. Thus, the anomalous SSH crosses the SST front between the J1 and J2, where the north–south shift of the front is large as depicted by the OEI (Figs. 1a and 2a). In addition, the SSH anomaly tends to correlate negatively with the IJI in the northwestern box and farther to the northeast. These relations imply, by geostrophy, that anomalous northeastward cross-frontal flow occurs when the IJI is large. This is consistent with the relation between the J1 strength and the SST anomaly shown in Fig. 2a. Indeed, a composite analysis (Supplementary Figure 2) also indicates that the anomalous heat transport by the J1 contributes to substantial warming of the SAFZ in winter, and causes the upward heat flux from ocean to atmosphere over the J1.

**SSH signals of subtropical origin and subarctic origin**. Now, a focal question is how such a stationary northward jet can form away from the western boundary. In previous studies, it was suggested that the J1 flows along the eastern flank of bottom rise ranging from **A** to **A′**[9,11,14] (see Fig. 1b). We note that the height of the rises **A-A′** is low (~500 m) compared with the depth of the North Pacific Ocean, and therefore, one may suspect that bar-oclinic Rossby waves would ignore such a low topographic feature easily[19].

As Fig. 2b indicates, however, the decadal variations of the SSH in the southeastern box (SSH-S) are obviously different from the SSH in the northwestern box (SSH-N) even though these boxes are only 100 km apart from each other. Correspondingly, the spatial pattern of simultaneously correlated SSH with respect to the decadal SSH-S (indicated as lag = 0.0 year in Fig. 3a) is different from that with the SSH-N variations (lag = 0.0 year in Fig. 3b). That is, the SSH anomaly correlated with the SSH-S extends northeastward, while the SSH anomaly correlated with the SSH-N is confined in the northwestern subarctic. Since the amplitude of the decadal variations of SSH-S is larger than that of the SSH-N (Fig. 2b), the IJI's decadal variations are largely determined by the SSH-S.

A lag correlation with respect to the SSH-S exhibits westward propagation of the SSH anomaly (Fig. 3a), in which the anomaly appears around 170°E with 3 years lead. The propagation speed is evaluated as ~0.015 ms$^{-1}$ if we consider the SSH signal moves from 170°E to 155°E in 2 years, consistent with the phase speed of baroclinic Rossby waves of 41°N[4,10,18]. The SSH anomaly then enters the southeastern box during the period between year −1 and year 0. The westward propagation is blocked at the J1, as the

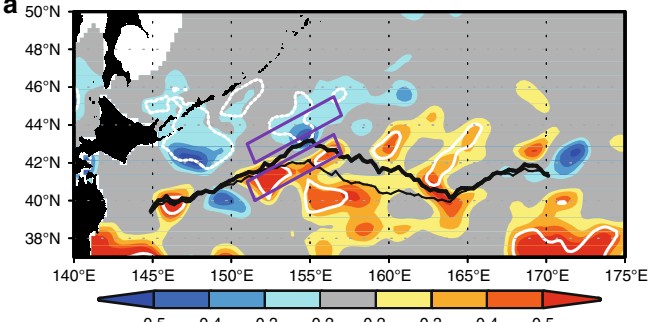

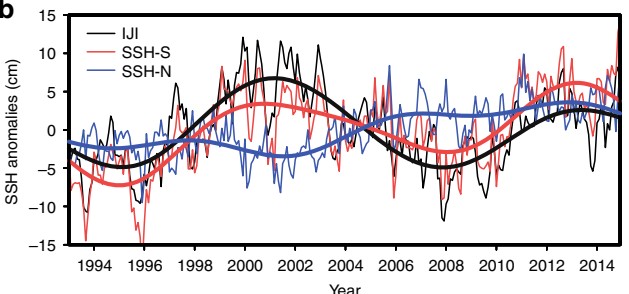

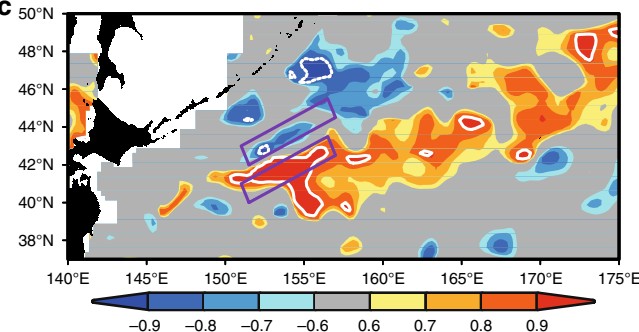

**Fig. 2** Isoguchi Jet strength. **a** Regression map (shade; °C) between SST anomaly (December, January, February) derived from OISST[20] and normalized time series of Isoguchi Jet Index (IJI), where the normalization is made with respect to its standard deviation. Here the IJI is defined by the difference between the satellite derived SSH anomaly[45] in the southeastern box and that in the northwestern box, where definition of each box is given in Methods. White contours denote significance level greater than 95%. Solid lines denote the SST front latitude $\overline{\Phi}(x) + \mathrm{EOF1}(x) \cdot \sigma$ derived from OISST, where $\sigma = 0$ (thin line) and $\sigma = 1$ (thick line). **b** Monthly time series of IJI (black), SSH anomaly in the southeastern box (SSH-S; red) and that in the northwestern box (SSH-N; blue). Thick lines denote 7-year low-pass filtered time series. **c** Correlation map between low-pass filtered (>7 years) SSH anomaly and normalized IJI time series. White contours denote significance level greater than 95%

SSH anomaly does not propagate further westward (Supplementary Figure 3). This is also consistent with previous results by a long baroclinic Rossby wave model[20], where the modeled SSH north of 40°N represents observed SSH well up to 160°E, but loses correlation quickly further westward.

As for the decadal SSH-N, we have found that the SSH anomaly that originates in the northern part of the western subarctic gyre propagates southwestward and enters the north-western box (Fig. 3b); the SSH anomaly likely follows the eastern flank of the rises **A-A′**. It takes 3 years to reach the northwestern box, and therefore, the phase speed is evaluated as ~0.007 m s$^{-1}$. This implies that the anomalous SSH originating in the north

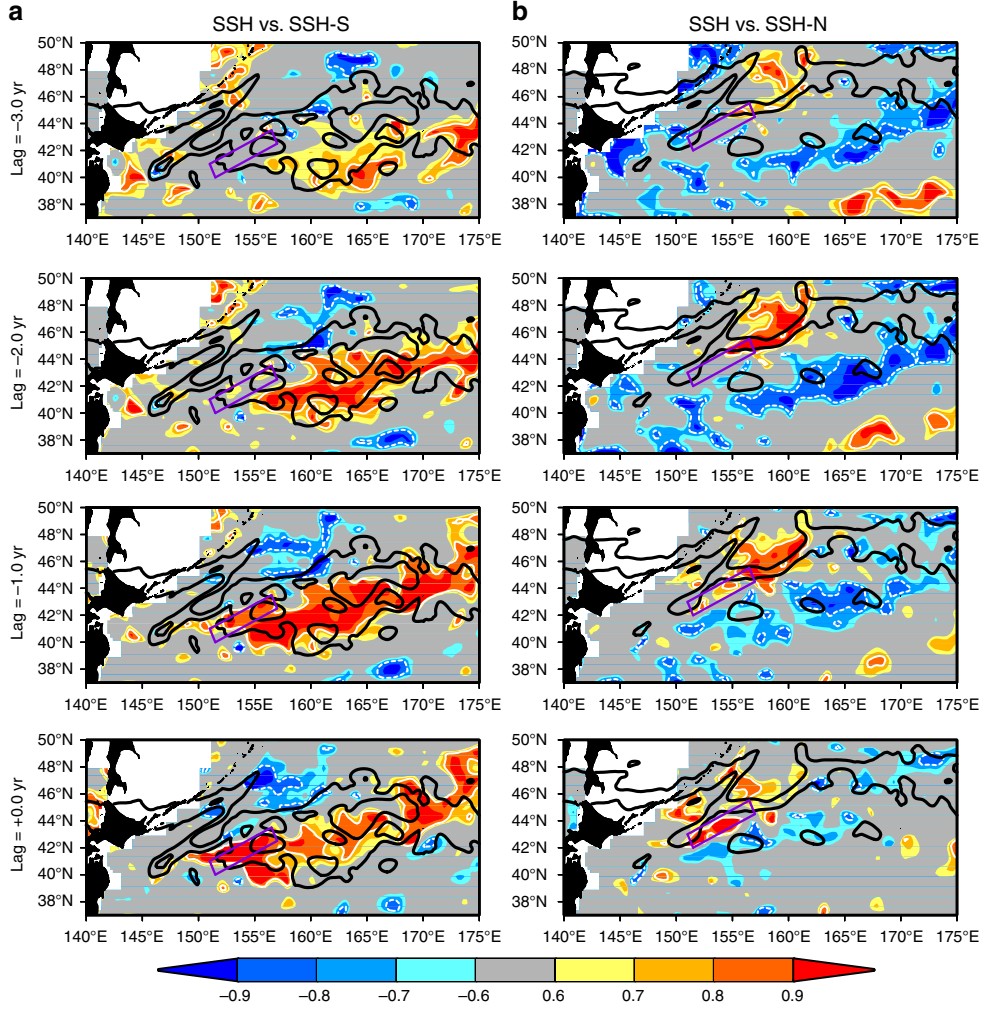

**Fig. 3** Propagation of low-frequency signals of the subtropical origin and the subpolar origin. Lag correlation map between low-pass filtered (>7 years) SSH anomaly from satellite[45] and normalized SSH time series in the **a** southeastern box (SSH-S), and **b** the northwestern box (SSH-N). The southeastern box is indicated by a purple box in **a**, while the northwestern box is in **b**. Lag = 0.0 year denotes the simultaneous correlation. Lag = −3.0 year in **a** denotes that the SSH anomaly leads the SSH-S time series by 3 years. White contours denote significance level greater than 95%. Black contours represent the characteristic curves $Q_c$ given by Eq. (2) with **a** 4.5 and $8 \times 10^{-6}\,\mathrm{s}^{-1}$, and **b** 8 and $12 \times 10^{-6}\,\mathrm{s}^{-1}$

must be baroclinic, because barotropic topographic Rossby waves should propagate much faster (>10 m s$^{-1}$).

The above SSH analysis indicates that the baroclinic Rossby waves of two different origins—the subtropical gyre to the east (subtropical origin) and the western subarctic gyre to the north (subarctic origin)—converge at the J1 and causes the decadal variations as shown in Fig. 2b. This gives an implication for the formation of the quasi-stationary baroclinic jet as well. That is, the stationary Isoguchi J1 may form at a location where steadily forced baroclinic Rossby waves with a thick upper layer of subtropical origin and a thin layer of subarctic origin encounter each other.

**Baroclinic Rossby wave characteristics**. The above propagation features of decadal signals motivate us to re-consider the baroclinic-Rossby-wave characteristics[21–23] over bottom topography (see Methods; also see Miyama et al.[24] including detailed scaling arguments). We hypothesize that the characteristics are deflected along the sea-floor rises **A-A′**, so that the baroclinic Rossby waves of the subarctic origin propagate southwestward and encounter the westward propagating waves of the subtropical origin, as the decadal SSH anomalies do. Such a deflection of the characteristics from zonal lines of the ambient planetary vorticity

can be caused by the effects of a barotropic flow[25]. To illustrates this hypothesis, we consider a quasi-geostrophic potential vorticity equation in a rigid-lid 2-layer ocean with bottom topography. The evolution equation for the upper layer thickness $h$ over bottom topography is reduced to (see Methods)

$$\frac{\partial h}{\partial t} + C_x \frac{\partial h}{\partial x} + C_y \frac{\partial h}{\partial y} = \text{forcing}, \tag{1}$$

where $t$, $x$, and $y$ denote time, zonal position, and meridional position, respectively, and $(C_x, C_y)$ denotes the phase velocity of baroclinic Rossby waves. $(C_x, C_y)$ may be written in terms of the baroclinic Rossby wave characteristics $Q_c$, where

$$Q_c = \beta y + \frac{\psi_{\mathrm{Ta}}}{R_d^2}, \tag{2}$$

such that

$$(C_x, C_y) = \left( -\frac{\partial R_d^2 Q_c}{\partial y}, \frac{\partial R_d^2 Q_c}{\partial x} \right),$$

where $\beta$ is the meridional gradient of the Coriolis parameter $f$, i.e., $\beta = \frac{df}{dy}$, $R_d$ denotes the internal radius of deformation, and $\psi_{\mathrm{Ta}}$

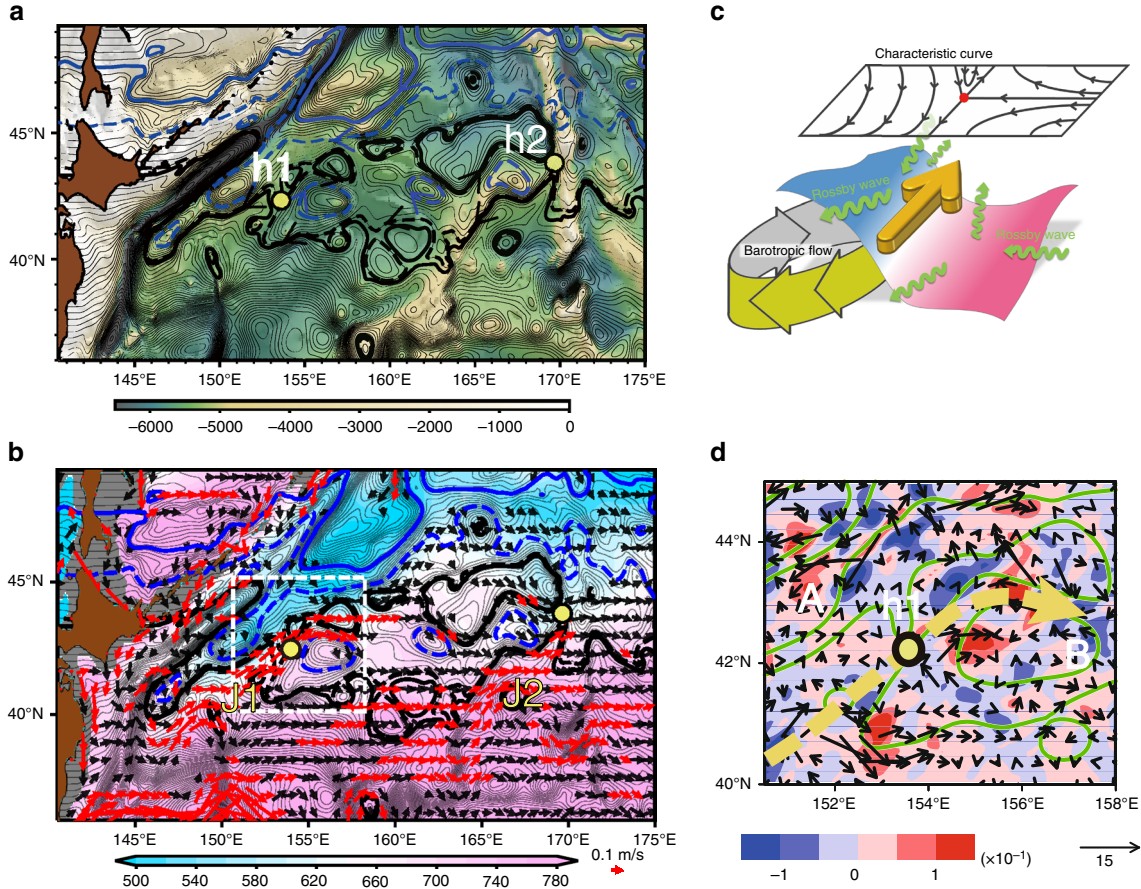

**Fig. 4** Surface jet formation and baroclinic Rossby wave characteristic curves. **a** Characteristic curves and bottom topography. Contours represent $Q_c$ given by Eq. (2) evaluated in the western North Pacific. The hyperbolic stationary points h1 and h2 associated with J1 and J2, respectively, are denoted by yellow dots. Contour interval is $0.5 \times 10^{-6}\,\mathrm{s}^{-1}$, where thick lines are 4.5 (solid black), 5.0 (dashed black), 8.0 (dashed blue), 12.0 (solid blue) $\times 10^{-6}\,\mathrm{s}^{-1}$. Shade represents bottom topography. **b** Characteristic curves, interfacial depth, and surface current. Contours are $Q_c$, which is the same as those in **a**. Shade represents isopycnal depth (m) of $\sigma_\theta = 27.2$ derived from JCOPE2 ocean reanalysis. Red (black) arrows denote the surface velocity faster than $0.1\,\mathrm{m\,s}^{-1}$ (between $0.1\,\mathrm{m\,s}^{-1}$ and $0.05\,\mathrm{m\,s}^{-1}$) from JCOPE2. Yellow dots denote the hyperbolic stationary points h1 and h2. **c** Schematic plot representing the relation between characteristic curves and surface jet formation. Shaded distorted plane represents the depth of an isopycnal surface. **d** Eddy flux (vectors; $\mathrm{m}^2\,\mathrm{s}^{-1}$) adjacent to the J1 and its convergence/divergence (color shade; $\mathrm{m\,s}^{-1}$) inside the dashed box in **b**, where convergence is defined to be positive. Specific formulae of the eddy flux are given by Eq. (13) in Methods. Eddy thickness flux, which is the first term of Eq. (13), dominates over other terms of a factor of $10^2$. Green contours denote $Q_c$ with 5.0, 8.0 ($\times 10^{-6}\,\mathrm{s}^{-1}$). Yellow dot denotes h1, and yellow arrow denotes a schematic J1 path. Rises **A** and **B** are indicated

denotes the ambient barotropic streamfunction driven by external forcing such as wind and eddy-topography interaction. A derivation of Eqs. (1) and (2), as well as a specific form of the forcing term on the right-hand side of Eq. (1), is given in Methods. Here we consider $\psi_{\mathrm{Ta}}$ to be a given parameter, and thus a set of Eqs. (1) and (2) is regarded as a diagnostic tool for the baroclinic wave propagation for a given $\psi_{\mathrm{Ta}}$.

Equation (2) indicates that $Q_c$ comprises not only planetary waves that propagate westward associated with $\beta$ but also advection by the barotropic flow $\psi_{\mathrm{Ta}}$. Note that topographic terms are not present explicitly in Eq. (2) although $Q_c$ is evaluated over bottom topography. This is because baroclinic Rossby waves tend to be surface trapped in the presence of topography[26], that is, the upper-layer motion and the lower-layer motion are decoupled, which makes baroclinic Rossby waves insensitive to bottom topography[19].

We evaluate $Q_c(x,y)$ utilizing the barotropic flow field of JCOPE2[13], which reproduces the Isoguchi Jets and the SST-front structure well (Fig. 1a). $\psi_{\mathrm{Ta}}$ is evaluated by a vertically averaged velocity of JCOPE2. $R_d$ is set at 27.4 km based on Killworth et al.[27] so that unperturbed westward phase speed is evaluated as

$1.3 \times 10^{-2}\,\mathrm{m\,s}^{-1}$ at $41°\mathrm{N}$[10], where $\beta = 1.72 \times 10^{-11}\,\mathrm{m}^{-1}\,\mathrm{s}^{-1}$. Figure 4a indicates that in general characteristic curves are deformed from zonal lines that represent the westward propagation of planetary baroclinic Rossby waves. Substantial deformation occurs over topographic features (Fig. 4a) where the barotropic transport is large (Fig. 1b). Therefore, although Eq. (2) does not include the topographic term explicitly, the baroclinic Rossby waves feel topography through the barotropic flow term in $Q_c$.

A closer examination around the J1 indicates that the characteristic curves tend to be clockwise over the rises **A-A′** (Fig. 4a) which are attributed to the clockwise barotropic flows over the rises **A-A′** (Fig. 1b). This barotropic flow is likely induced by the interaction between eddies and topography[28–31]. Similarly, clockwise characteristic curves located over the rise B where the barotropic flow is present (Fig. 1b). Therefore, the barotropic flow is essential for $Q_c\,(x,y)$ over these topographic features adjacent to the J1.

To understand the formation mechanism of the J1 based on the above $Q_c$ distribution, let us focus here on the hyperbolic point h1 ($42.5°\mathrm{N}$, $153.5°\mathrm{E}$). As indicated by the clockwise characteristic

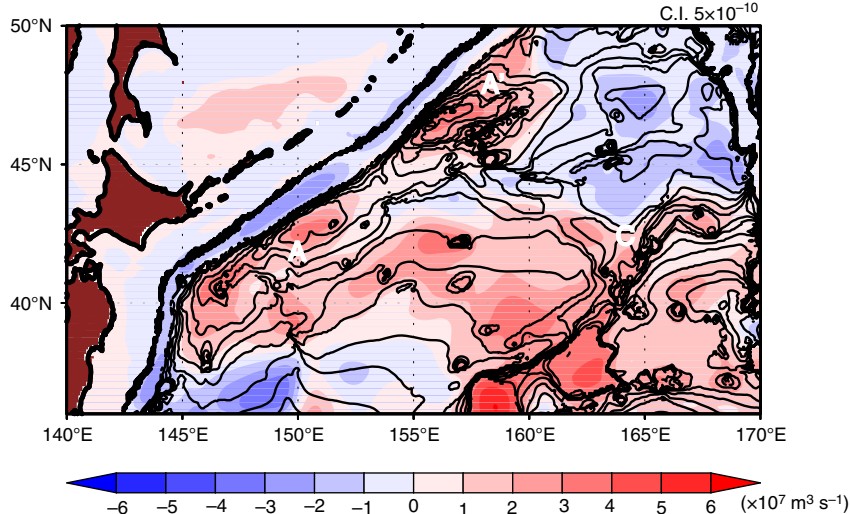

**Fig. 5** Barotropic ambient potential vorticity and barotropic streamfunction. Contour lines denote barotropic ambient potential vorticity $f/H$ (contour interval: $5 \times 10^{-10}\,\mathrm{s}^{-1}\,\mathrm{m}^{-1}$) and shade denotes streamfunction derived from the JCOPE2 reanalysis. Closed contours of $f/H$ are present over the rises **A**, **A′**, and **C**

curves around the rises **A**-**A′**, baroclinic Rossby waves propagate southwestward on the southeastern flank of the rises, bringing the thin upper layer originating in the subarctic region (blue region in Fig. 4b) down to ~41°N. The characteristic curves representing the westward propagation of subtropical origin (pinkish region in Fig. 4b), on the other hand, are shown to deflect northward on the western flank of the rise B and brings the thick layer toward h1. This convergence of characteristic curves in the vicinity of h1, therefore, creates a thickness jump, and hence a geostrophic jet corresponding to the J1 (Fig. 4b). Figure 4c provides a schematic summary of this process.

The convergence of characteristic curves above has a close relationship with the propagation of the decadal SSH anomaly from satellite depicted in Fig. 3. As Fig. 3b indicates, the SSH anomaly of the subarctic origin moves southwestward following the $Q_c$ contours from lag = −3 year to −1 year, and finally enters the northwestern box at lag = 0. The westward propagation of the decadal SSH anomaly of subtropical origin (Fig. 3a) also closely follows the $Q_c$ contours from lag = −3 year to −1 year, enters the southeastern box at lag = −1 year, and is finally blocked at lag = 0 year. The blocking of the westward propagation corresponds to the convergence of characteristic curves at the hyperbolic point h1 as depicted in Fig. 4c.

The above analysis implies that $Q_c$ is a useful tool to diagnose the evolution of upper layer thickness. According to Eq. (2), therefore, deflection of the baroclinic Rossby wave propagation from due west may well be caused by the ambient barotropic flow $\psi_{\mathrm{Ta}}$. This leads to a conclusion that the northeastward quasi-stationary jet formation away from the western boundary is attributed to the barotropic flows generated over the small-amplitude rises **A**-**A′** and **B**.

Another effect that possibly affects the J1 formation is eddy fluxes in Eq. (13) in Methods. This causes the upper layer thickness flux across $Q_c$ contours, in contrast to the along-$Q_c$ thickness flux by baroclinic Rossby waves. Here we focus on this effect inside the box of Fig. 4b adjacent to the J1. As shown by Fig. 4d, the eddy thickness flux is convergent (the layer thickening) approximately at 41°N, 153°E and 42.5°N, 155°E eastward of the J1. This causes deepening of the upper layer around the rise **B**. On the other hand, the eddy flux is divergent (the layer thinning) approximately at 43.5°N, 155°E northeastward of h1. Since eddy flux vectors go across a $Q_c$ contour northward from this region to the subarctic gyre (the blue region

in Fig. 4b), this thinning is likely caused by the southward eddy flux of the thin upper layer from the subarctic gyre. While baroclinic Rossby waves may not enter this region because of a closed $Q_c$ contour, the eddy thickness flux likely causes the layer thinning instead. The eddy thickness flux thus tends to enhance the J1, since the flux causes thickening (thinning) in the northern (southern) side of the jet.

**Barotropic flow over small-amplitude rises and jet formation.** The above analysis indicates that the clockwise barotropic flow over the rises **A**-**A′** and the rise **B** in Fig. 1b is essential for the convergence between the thin subarctic layer and the thick sub-tropical layer, and consequently for the jet formation (Fig. 4b). Since the rise has small amplitude so that quasi-geostrophic dynamics are valid, we expect that a classical theory on geostrophic turbulence may well apply, in which eddies diffuse anomalous potential vorticity associated with topography while conserving energy[30,32]. In particular, strong barotropic flows may be generated if closed contours of $f/H$ are present, where $H$ is the ocean depth, because potential vorticity tends to be homogenized inside the closed contours. This eddy-topography interaction likely generates the clockwise (or, more precisely, anticyclonic) barotropic currents over the rises **A**-**A′** since the $f/H$ contours there are closed (Fig. 5). Closed $f/H$ contours can readily form in the high-latitude ocean even with such a small-amplitude topo-graphy with a height of 500 m because $\beta$ decreases and $f\partial H/\partial y$ increases (for a given $\partial H/\partial y$) with latitude.

To illustrate the effects of topography, we conducted an idealized two-layer model simulation with a simple topographic feature (Fig. 6), where the model basin has a depth of 5500 m with a sea-floor elevation of 500 m. The model is forced by wind whose Sverdrup transport function is shown in Fig. 6a. The topography is located across the boundary between the subtropical and subpolar gyres like the rises **A**-**A′**. Detailed configurations of the model are presented in Methods.

As expected, a northeastward surface jet forms over the eastern flank of the topography in a similar manner to the Isoguchi J1 (Fig. 6b). A thin upper layer of subarctic origin intrudes southward over the eastern flank of the topography as indicated by deformed characteristic curves (yellow contour in Fig. 6b). At the same time, a characteristic curve originating in the subtropical gyre, like a light-blue contour, extends

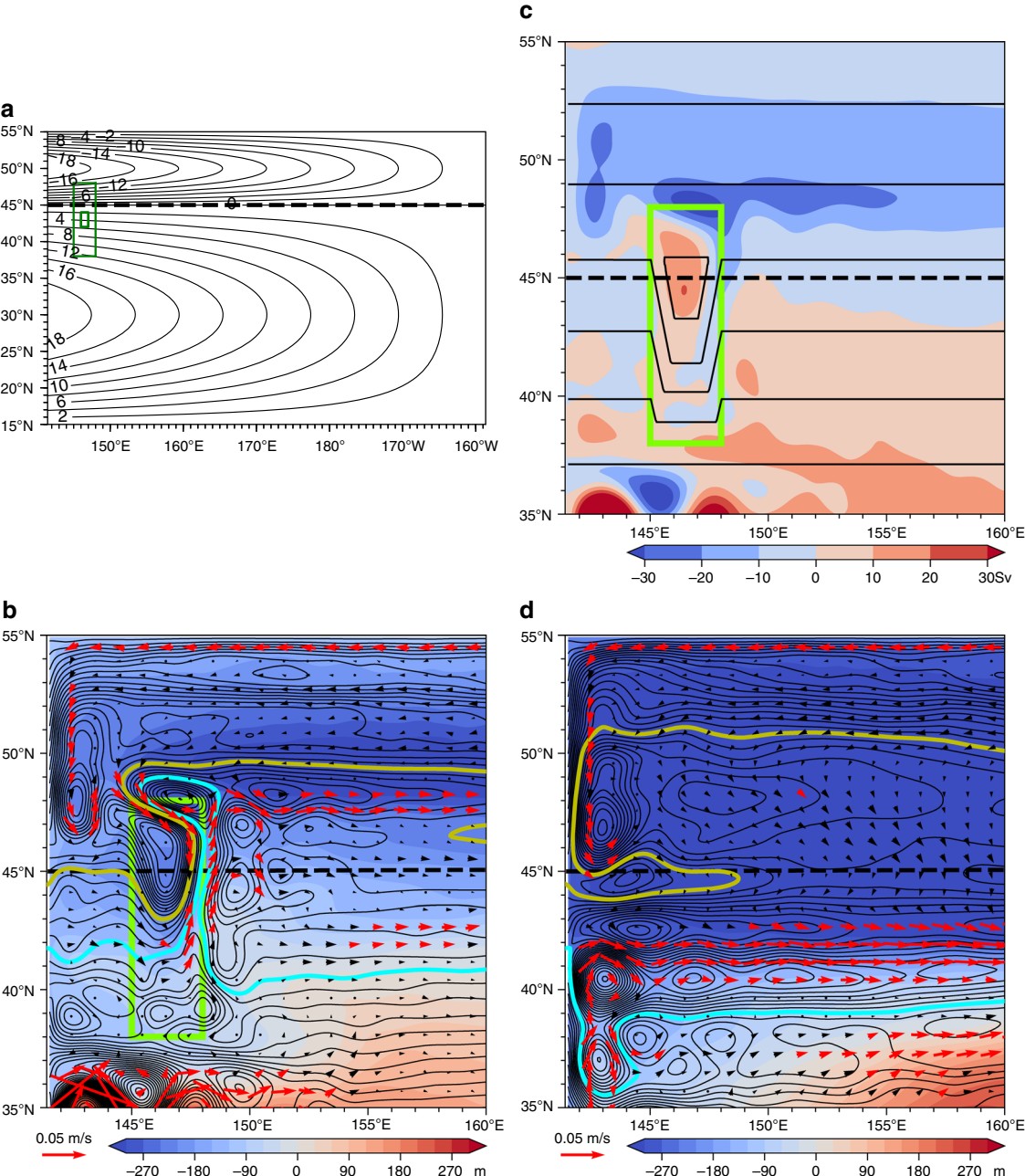

**Fig. 6** Idealized two-layer simulation. **a** Model basin with a depth of 5500 m. The bottom topography with a height of 500 m is depicted by green lines denoting ocean depth contours of 5500 m and 5000 m. Solid contours denote wind-driven transport function (×10⁶ m³ s⁻¹). **b** Bottom Topography Case. Upper layer thickness anomaly (color shade; m), upper layer velocity (arrow) and characteristic curves $Q_c$ with a contour interval of $0.5 \times 10^6$ m³ s⁻¹ are shown. Green box denotes the location of bottom topography. **c** Barotropic streamfunction (shade ×10⁶ m³ s⁻¹) and $f/H$ (contour interval $0.1 \times 10^{-8}$ m⁻¹ s⁻¹) for Bottom Topography Case. Green box indicates the location of bottom topography. **d** Same as **b** but for No Bottom Topography Case

northward in the east of the topographic feature. This resembles the results of the JCOPE2 in the preceding section (Fig. 4b). The deflection of the characteristic curves $Q_c$ is attributed to the clockwise barotropic streamfunction over topography (Fig. 6c). Strong barotropic currents are generated by eddy-topography interaction over a topographic rise that has closed $f/H$ contours, even though the rise is as high as 500 m[24]. If we remove the topography, then the northward surface jet away from the western boundary disappears (Fig. 6d). Therefore, the barotropic flow produced over the low topographic rise is essential for the formation of the surface stationary jet corresponding to the J1.

The discussion above is consistent with a previous modeling result that the J1 is reproduced only when the model's horizontal resolution is high enough so that eddy-topography interaction is properly represented[9]. Indeed, the clockwise barotropic circulation in the subarctic gyre over the rise A is also seen in other models, such as the community ocean model of the Meteorological Research Institute (MRI.COM; see Methods)[33,34], where the Isoguchi J1 is also reproduced (Fig. 7). Consequently, the J1 formation is coupled tightly with the barotropic flow over bottom topography.

It is difficult to measure barotropic flows in the real ocean. In particular, the clockwise barotropic circulation over the rise A-A′

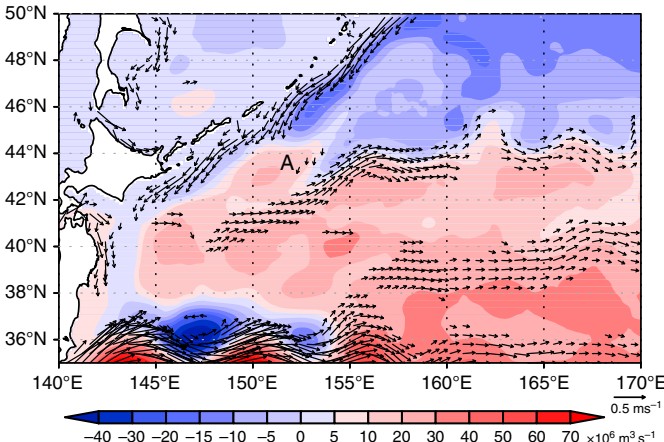

**Fig. 7** Barotropic streamfunction and a jet formation in a western North Pacific simulation. Temporal mean field of barotropic streamfunction (color shade; ×10⁶ m³ s⁻¹) and surface current at a depth of 12 m (vector, cm s⁻¹) of the Meteorological Research Institute Community Ocean Model (MRI. COM)[32,33] are indicated. The simulation exhibits anticyclonic barotropic flow over the rise **A** and the Isoguchi J1. Model configurations are described in Methods

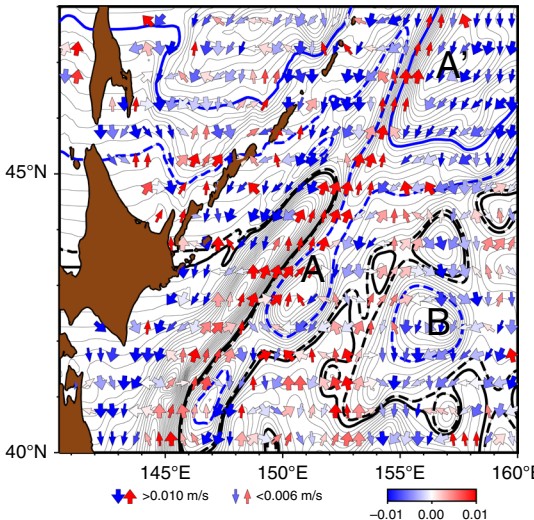

**Fig. 8** Implication of meridional barotropic flows over topography by tracking eddy propagation. Vectors denote eddy propagation velocity derived by tracking eddies retrieved from satellite SSH[53] (see Methods). Color scale indicates the meridional propagation speed, where red (blue) denotes northward (southward). Characteristics $Q_c = \beta y + \frac{\psi_{Ta}}{R_d^2}$, as well as analyses of eddy propagation from satellite SSH[34], imply that meridional propagation of eddies may be caused by ambient barotropic flows. Contours denote $Q_c$ (contour interval 0.5 × 10⁻⁶ s⁻¹) with thick lines of 4.5 (solid black), 5.0 (dashed black), 8.0 (dashed blue), and 12.0 (solid blue) × 10⁻⁶ s⁻¹. Topographic rises **A**, **A′**, and **B** are indicated

was not reported in the literature. Nevertheless, judging from the southward propagation of the decadal SSH anomaly (Fig. 3b), the clockwise barotropic flow likely exists. To bolster this further, we present the propagation of eddies by tracking satellite-derived SSH anomalies (see Methods for eddy tracking). As shown in Fig. 8, the eddy propagation direction rotates greatly from due west over the rises **A-A′**. Eddies move southwestward (northeastward) over the eastern (western) flank of the rise, exhibiting the clockwise propagation. This is in contrast to an eddy

propagation feature in the world ocean[35]; the meridional deflection of eddy propagation is small—only 1.5° of rotation in propagation direction from due west—even if eddies are strongly non-linear. It was thus suggested that meridional propagation of eddies would be an effect by ambient flows such as mean barotropic currents[35]. The propagation of eddies in Fig. 8 therefore indicates the presence of ambient clockwise barotropic flows, although these were overlooked previously as they are apparently masked by the wind-driven surface circulation which is anticlockwise[36]. In the SAFZ, further, eddies on the western flank of the rise **B** propagate northward toward the confluence h1, consistently with the characteristic curves. In conclusion, the observed eddy propagation supports the presence of clockwise barotropic currents over the rises **A-A′** and the rise **B** which are essential for the J1 formation.

**Formation of the Isoguchi J1 and SST front in the SAFZ.** In this study, we have shown that decadal SST variations in the SAFZ are closely related to the strength of a quasi-stationary jet, the Isoguchi J1, which advects heat northeastward. A focal question here is how such a quasi-stationary jet forms away from the western boundary, and why it is collocated with the small-amplitude rises even though their elevations (~500 m) are much smaller than the depth of the North Pacific Ocean. We have shown that the small-amplitude rises anchor the surface jet effectively through barotropic flows driven by eddy-topography interaction. These barotropic flows deform the characteristic curves of baroclinic Rossby waves greatly, which in turn brings a thin upper layer in the subarctic gyre southward and creates a layer-thickness jump at which the westward propagation of the thick upper-layer signal, originating in the subtropical gyre, is blocked (Supplementary Figure 4). Since the Isoguchi J1 is located in the transition area between the subtropical and subarctic gyres, it transports heat northward and produces strong SST fronts, thereby likely imposing substantial impacts on atmospheric circulation and climate system in the Northern Hemisphere as shown in the previous studies[1-7]. Further, the jet also has remarkable effects on the inter-gyre exchange of fresh water and materials such as biogeochemical nutrients in the North Pacific Ocean[37].

**Discussion**
So far, we have focused on the J1 formation processes. The J2 likely forms in a similar manner associated with the rise **C**, including a ridge-like feature extending southwestward (Fig. 1a, b), whose amplitude is approximately 500 m. A clockwise barotropic flow is generated over the rise **C** (Fig. 1b) collocated with closed contours of $f/H$ (Fig. 5). This deforms the characteristic curves (Fig. 4a), which bring a thin layer from the hyperbolic point h2 southwestward along the ridge up to approximately 40° N, 164°E (Fig. 4b). The southwestward deflection of the baroclinic Rossby wave propagation between 170°E and 164°E is also captured in the satellite decadal SSH signal (for example, lag = −2 year in Fig. 3a), which closely follows the characteristic curve that goes out southwestward from h2 (Fig. 4a). Since a thicker layer of the subtropical origin is observed eastward of this characteristic curve (Fig. 4b), a layer-thickness jump occurs along the eastern flank of the rise **C**, thereby yielding the J2 formation. The formation mechanism of the J2 is the same as the formation mechanism of the J1, although the recirculation gyres beside the J2 also affect its path. As indicated by Supplementary Figure 5, both J1 and J2 exhibit surface-intensified structure on the eastern flank of topography, consistent with the convergence of characteristic curves of baroclinic Rossby waves.

The sensitivity of the surface currents in the subarctic ocean to small-amplitude topographic features urges us to examine bathymetric data carefully in applying them to ocean models

because in situ acoustic bathymetric measurements are still sparse. Sensitivity experiments using various bathymetry datasets may be useful as the model resolution increases; otherwise small-amplitude seafloor elevations could be overlooked. Improvement of bathymetric dataset is on-going by adding estimation of the seafloor elevations from satellite SSH as well as adding more in situ acoustic data[38].

The theoretical framework presented in this study may be applicable to other subpolar oceans where barotropic flows are strong enough. A prominent example is an anticyclonic barotropic flow over the Zapiola Rise in the South Atlantic whose transport exceeds 100 Sv (1 Sv = $10^6$ m$^3$ s$^{-1}$)[39,40]. The height of the rise is about 1000 m in the ocean with a depth of ~5800 m. The Zapiola Rise likely imposes remarkable impacts on the surface currents and fronts in the Brazil and Malvinas Confluence Region[39]. Further, since this area is located between the Atlantic Ocean and the Antarctic Circumpolar Current (ACC) in the Southern Ocean, the rise is likely to have significant influences not only on SST and SST fronts but also on mixing and water mass transformations between these oceans. Another example is the ACC in the Pacific sector. Recent modeling studies showed that the location of the ACC fronts in the Pacific sector was unchanged in response to wind-forcing changes in climate change simulations, although the topography there is almost flat[41]. However, on closer examination there are large scale closed $f/H$ contours in the Pacific sector of the Southern Ocean[42]. This suggests that the surface fronts of the ACC there could be anchored to bottom topography because barotropic flows may be controlled, at least partly, by the closed $f/H$ contours. As such, this study reveals efficient effects of small-amplitude topography on regulating surface jets and fronts in the subpolar oceans, where frontal scale SST variations likely induce far-reaching atmospheric responses through active storm tracks[1–7,43,44].

## Methods

**Data**. JCOPE2 is an ocean reanalysis product in which the satellite-derived SSH anomaly, satellite-derived SST, and vertical profile of potential temperature and salinity are assimilated to an ocean general circulation model using a three-dimensional variational method[13]. Reanalysis is made during the period 1993–2013. The ocean model is based on the Princeton Ocean Model with a horizontal resolution of 1/12°. SST and surface velocity displayed in Fig. 1a are derived from JCOPE2. The barotropic current field for the characteristic curves in Fig. 3a and b was calculated using the JCOPE2 output.

Isoguchi Jet Index (IJI) was derived from the merged products of monthly mean SSH anomaly of Topex/Poseidon, Jason-1, and European Research Satellite altimeter observations from 1993 to 2014 by the French Archiving, Validation, and Interpolation of Satellite Oceanographic Data (AVISO) project with a spatial resolution of 1/3°[45,46]. The northwestern box is defined by (42.0°N, 151.5°E), (44.5°N, 157.0°E), (45.5°N, 156.5°E), (43.0°N, 151.0°E), whereas the southeastern box is defined by (40.0°N, 151.5°E), (42.5°N, 157.0°E), (43.5°N, 156.5°E), (41.0°N, 151.0°E). To extract the thermosteric signal in SSH anomaly caused by local atmospheric heat flux, we calculated the thermosteric signal from net surface heat flux data as follows[47]:

$$\frac{\partial \eta'(t)}{\partial t} = \frac{\alpha}{\rho_0 c_p}\left\{Q(t) - \overline{Q(t)}\right\}, \tag{3}$$

where $\rho_0$ is the reference density, $c_p$ is the specific heat of sea water, and $\alpha$ is the thermal expansion coefficient. The thermal expansion coefficient was calculated from the temperature and salinity averaged over the mixed layer based on the World Ocean Atlas 2005. $Q(t)$ is the net surface heat flux derived from the climatological monthly means of the NCEP-CFSR reanalysis data[48] from 1979 to 2014. The overbar denotes the annual average of the climatological monthly means from January to December.

SST correlation and regression with respect to IJI (Fig. 2a) was derived from the Optimum Interpolation Sea Surface Temperature (OISST) version 2[17] during 1993–2014. The OISST is an analysis constructed by combining observations from satellite, in situ and buoys on a global 1/4 degree grid. The correlation was made based on monthly mean data.

The significance of correlation is estimated by a Monte Carlo technique, using a phase randomization technique generating 1000 surrogate time series[49]. First, absolute Fourier amplitudes (square root of spectra) for the J1 index are estimated, and then 1000 surrogate time series are generated by an inverse Fourier transform with the observed Fourier amplitudes and randomized phases. Surrogate correlation coefficients are estimated between the surrogate IJI time series and each

variable (e.g., SSH anomaly). The relative position of the absolute value of the observed correlation coefficients in the sorted absolute values of the surrogate correlation coefficients gives the level of confidence for the observed correlation coefficient. Since this method takes into account the spectrum structure of the time series, this is more conservative rather than the significance test based on $t$-test.

**Formulation of the baroclinic Rossby wave characteristics**. We consider a quasi-geostrophic potential vorticity equation in a two-layer ocean with the upper and lower layer thicknesses $H_1$ and $H_2$, respectively, and the total thickness $H_T = H_1 + H_2$. The quasi-geostrophic streamfunctions for the upper and the lower layer are $\psi_1$ and $\psi_2$, respectively. Evolution of low-frequency, long baroclinic Rossby waves in the presence of bottom topography may be written by the following coupled equations[24]:

$$-\frac{\partial}{\partial t}\frac{\psi_c}{R_d^2} + J(\psi_c, \hat{Q}_c) - J\left(\psi_T, \frac{f_0 Z_b}{H_2}\right) = \frac{f_0}{H_1} w_E + \Im_c, \tag{4}$$

$$J\left(\psi_T, \beta y + \frac{f_0 Z_b}{H_T}\right) - \frac{H_1}{H_T}J\left(\psi_c, \frac{f_0 Z_b}{H_T}\right) = \frac{f_0}{H_T} w_E + \Im_T, \tag{5}$$

where $\psi_c = \psi_1 - \psi_2$, $H_T \psi_T = H_1 \psi_1 + H_2 \psi_2$, and

$$\hat{Q}_c = \beta y + \frac{\psi_T}{R_d^2} + \frac{H_1}{H_2}\frac{f_0 Z_b}{H_T}.$$

Here, $R_d$ is the internal radius of deformation, $f_0$ is the Coriolis parameter on the reference latitude, $Z_b$ is the height of the bottom topography, and $w_E$ denotes the Ekman pumping velocity. $J$ is the Jacobian, and $\Im_c$ and $\Im_T$ denote eddy forcing terms which will be specified later. Since $\psi_c$ is proportional to the thickness anomaly of the upper layer, Eq. (4) represents the propagation of baroclinic Rossby waves. Note that, although our purpose is to discuss a mean field of a thickness front, the time derivative is retained in Eq. (4) for the sake of physical interpretation from a point of view of baroclinic Rossby waves. The third term on the left-hand side (LHS) yields a modification through the barotropic term $\psi_T$ given by Eq. (5). The barotropic flows in Eq. (5) is driven by the wind and eddy forcing terms on the right-hand side (RHS). In addition, Eq. (5) includes a barotropic-baroclinic coupling term in the LHS through so-called the joint effect of baroclinicity and bottom relief (JEBAR). Propagation of baroclinic Rossby waves is modified by this JEBAR term, together with the third term on the LHS of Eq. (4).

In order to obtain $\psi_T$ specifically, we assume $\epsilon = H_1/H_2$ is a small parameter. In the subarctic Pacific Ocean with a depth of 6000 m, this assumption is reasonable as the depth of 27.2 $\sigma_\theta$ is approximately 600 m (Fig. 4b) and hence $\epsilon \approx 0.1$. Thus, we expand $\psi_T$ such that

$$\psi_T = \psi_{Ta} + \epsilon \psi_T^{(1)} + \dots . \tag{6}$$

Here $\psi_{Ta}$ is an ambient barotropic flow driven by wind and eddies, which satisfies

$$J(\psi_{Ta}, Q_T) = \frac{f_0}{H_T} w_E + \Im_T,$$

$$Q_T = \beta y + \frac{f_0 Z_b}{H_T},$$

where $Q_T$ is a quasi-geostrophic expression of $f/H$ in the text. $\psi_{Ta}$ deforms greatly around closed $Q_T$ contours since $\psi_{Ta}$ tends to be parallel to $Q_T$. Here we consider $\psi_{Ta}$ as a known parameter, derived from the known forcing $w_E$ and $\Im_T$. In other words, a set of Eqs. (4) and (5) is a diagnostic tool to evaluate characteristics of baroclinic Rossby waves with a given $\psi_{Ta}$ field.

The next order yields the barotropic flow production due to JEBAR such that

$$J\left(\psi_T^{(1)}, Q_T\right) = J\left(\psi_c, \frac{f_0 Z_b}{H_T}\right). \tag{7}$$

Suppose an arbitrary $\psi_c (x, y)$ that satisfies $\psi_c = 0$ at boundaries of a basin. Integrating Eq. (7) along an isoline of $Q_T = \beta y + \frac{f_0 Z_b}{H_T}$ from the boundary, we obtain a barotropic correction $\psi_T^{(1)}$ due to JEBAR such that

$$\psi_T^{(1)}(x,y) = \psi_c(x,y) - \int^s \left(\frac{1}{\beta}\frac{\partial Q_T}{\partial n}\right)^{-1}\frac{\partial \psi_c}{\partial x}ds', \tag{8}$$

where $s(n)$ is the coordinate parallel (perpendicular) to the $Q_T$ contour; $s(n)$ corresponds to $x(y)$ over a flat bottom.

If Eq. (6) is substituted into Eq. (4) and Eq. (8) is utilized, then the coupling term (the third term) on the LHS of Eq. (4) becomes

$$J\left(\psi_{\mathrm{T}}, \frac{f_0 Z_b}{H_2}\right) = J\left(\psi_{\mathrm{Ta}}, \frac{f_0 Z_b}{H_2}\right) + J\left(\epsilon\psi_{\mathrm{c}} - \epsilon\int^s \left(\frac{1}{\beta}\frac{\partial Q_{\mathrm{T}}}{\partial n}\right)^{-1}\frac{\partial\psi_{\mathrm{c}}}{\partial x}\,ds', \frac{f_0 Z_b}{H_2}\right)$$

$$= J\left(\psi_{\mathrm{Ta}}, \frac{f_0 Z_b}{H_2}\right) + J\left(\psi_{\mathrm{c}}, \epsilon\frac{f_0 Z_b}{H_2}\right) - \epsilon\frac{H_{\mathrm{T}}}{H_2}J\left(\psi_{\mathrm{c}} - \int^s\left(\frac{1}{\beta}\frac{\partial Q_{\mathrm{T}}}{\partial n}\right)^{-1}\frac{\partial\psi_{\mathrm{c}}}{\partial x}\,ds', \beta y\right) \quad (9)$$

$$= J\left(\psi_{\mathrm{Ta}}, \frac{f_0 Z_b}{H_2}\right) + J\left(\psi_{\mathrm{c}}, \epsilon\frac{f_0 Z_b}{H_2}\right) - \epsilon\frac{H_{\mathrm{T}}}{H_2}\left\|J(\psi_{\mathrm{c}}, \beta y)\right\|,$$

where a relation $\frac{f_0 Z_b}{H_2} = \frac{H_{\mathrm{T}}}{H_2}(Q_{\mathrm{T}} - \beta y)$ is used to derive Eq. (9). Comparing Eq. (9) with the $J(\psi_{\mathrm{c}}, \hat{Q}_{\mathrm{c}})$ on the LHS of Eq. (4), we retain the second term, $J\left(\psi_{\mathrm{c}}, \epsilon\frac{f_0 Z_b}{H_2}\right)$, of Eq. (9) so that this term can be as large as $\left\|J(\psi_{\mathrm{c}}, \beta y)\right\|$ for a steep topography, whereas the third term of Eq. (9) is dropped because it is O ($\epsilon$) with respect to $\left\|J(\psi_{\mathrm{c}}, \beta y)\right\|$. Equation (4) is thus reduced to, by considering Eq. (9),

$$-\frac{\partial}{\partial t}\frac{\psi_{\mathrm{c}}}{R_{\mathrm{d}}^2} + J(\psi_{\mathrm{c}}, Q_{\mathrm{c}}) = \frac{f_0}{H_1}w_{\mathrm{E}} + \Im_{\mathrm{c}} + J\left(\psi_{\mathrm{Ta}}, \frac{f_0 Z_b}{H_2}\right), \quad (10)$$

where

$$Q_{\mathrm{c}} = \hat{Q}_{\mathrm{c}} - \epsilon\frac{f_0 Z_b}{H_2}$$
$$= \beta y + \frac{\psi_{\mathrm{Ta}}}{R_{\mathrm{d}}^2}. \quad (11)$$

Equation (11) is equivalent to Eq. (2). The topographic term in $Q_{\mathrm{c}}$ is canceled by the barotropic correction due to JEBAR. The insensitivity of the phase speed of baroclinic Rossby waves to the bottom topography $Z_b$ arises as a result of the decoupling between upper and lower layer motions in the presence of topography[19,24].

Equation (10) may be written in terms of the upper layer thickness disturbance $h = \frac{f_0}{\gamma}\psi_{\mathrm{c}}$ ($\gamma$ denotes reduced gravity) such that

$$\frac{\partial h}{\partial t} + C_x\frac{\partial h}{\partial x} + C_y\frac{\partial h}{\partial y} = -\frac{H_2}{H_{\mathrm{T}}}w_{\mathrm{E}} - \nabla\cdot\mathcal{F}_{\mathrm{c}} - \frac{H_1}{H_{\mathrm{T}}}J(\psi_{\mathrm{Ta}}, Z_b) \quad (12)$$

where

$$(C_x, C_y) = \left(-\frac{\partial R_{\mathrm{d}}^2 Q_{\mathrm{c}}}{\partial y}, \frac{\partial R_{\mathrm{d}}^2 Q_{\mathrm{c}}}{\partial x}\right)$$

and

$$\mathcal{F}_{\mathrm{c}} = \overline{(\mathbf{u}_{\mathrm{T}}'h')} - R_{\mathrm{d}}^2\overline{(\mathbf{u}_{\mathrm{T}}'\nabla^2 h')} - \frac{f_0 R_{\mathrm{d}}^2}{\gamma}\overline{(\mathbf{u}_{\mathrm{c}}'\nabla^2\psi_{\mathrm{T}}')} - \frac{(H_2 - H_1)}{H_{\mathrm{T}}}R_{\mathrm{d}}^2\overline{(\mathbf{u}_{\mathrm{c}}'\nabla^2 h')}, \quad (13)$$

where prime denotes deviation from the time mean field, $\mathbf{u}_{\mathrm{c}}' = (u_{\mathrm{c}}', v_{\mathrm{c}}') = \frac{\gamma}{f_0}\left(-\frac{\partial h'}{\partial y}, \frac{\partial h'}{\partial x}\right)$ and $\mathbf{u}_{\mathrm{T}}' = (u_{\mathrm{T}}', v_{\mathrm{T}}') = \left(-\frac{\partial\psi_{\mathrm{T}}'}{\partial y}, \frac{\partial\psi_{\mathrm{T}}'}{\partial x}\right)$. Recall that our purpose is to discuss a mean field of a jet formation, and hence the deviation is defined with respect to the temporal mean field; the time derivative in Eq. (4) is retained for the sake of physical interpretation from a point of view of baroclinic Rossby waves. Here, the RHS in Eq. (12) gives a specific form of the forcing terms in Eq. (1) in the text. The eddy flux evaluated in Fig. 4d given by Eq. (13), using the depth anomaly of the 27.2 $\sigma_\theta$ of JCOPE2. Note that the first term on the RHS, representing the thickness flux, is dominant over other terms by a factor of $10^2$. Detailed derivation of Eq. (13) is given in Miyama et al.[24].

**Parameters for evaluating $Q_{\mathrm{c}}$.** The barotropic current field $\psi_{\mathrm{Ta}}$ (Fig. 1b) were calculated using the mean velocity fields of JCOPE2 during the period 1993–2013, by averaging over the whole depth. The baroclinic Rossby radius $R_{\mathrm{d}}$ estimated to be 27.4 km based on Killworth et al.[27]. $Q_{\mathrm{c}}$ in Fig. 4a and b is evaluated by Eq. (2) with these $\psi_{\mathrm{Ta}}$ and $R_{\mathrm{d}}$, together with $f_0 = 9.5\times10^{-5}\,\mathrm{s}^{-1}$ and $\beta = 1.7\times10^{-11}\,\mathrm{m}^{-1}\,\mathrm{s}^{-1}$ at 41°N. This yields the phase speed of the unperturbed baroclinic planetary Rossby waves to be $1.3\times10^{-2}\,\mathrm{m\,s}^{-1}$ at 41°N. Note that 10 Sv is equivalent to $2.0\times10^{-2}\,\mathrm{m\,s}^{-1}$ for a barotropic current with a width of 100 km and a depth of 5000 m. This implies that the baroclinic Rossby waves are readily affected by the barotropic flow in the subarctic ocean.

**Idealized two-layer model.** The HYbrid Coordinate Ocean Model (HYCOM)[50] in its isopycnal two-layer model[51] was used to conduct an idealized model depicted in Fig. 6. The basin covered the area 141.5°E–158.5°W by 15°N–55°N. The zonal grid spacing was 1/12°, and the meridional grid spacing was refined at higher latitudes to keep the grid cells square. The initial upper and lower layer thicknesses were 500 and 5000 m, respectively. A non-slip boundary condition is applied to the lateral boundaries. The potential densities $\sigma_\theta$ of the two layers were 26.7 and 27.65 based on the density structure around the Isoguchi J1. The model is driven by the wind

stress

$$\tau = \begin{cases} -\tau_1\cos\left(\pi\frac{\varphi-\varphi_1}{\varphi_2-\varphi_1}\right) & \varphi_1\leq\varphi<\varphi_2 \\ \tau_1-\tau_2+\tau_2\cos\left(\pi\frac{\varphi-\varphi_2}{\varphi_3-\varphi_2}\right) & \varphi_2\leq\varphi<\varphi_3 \end{cases},$$

where $\varphi$ is the latitude, $\varphi_1 = 15°\mathrm{N}$, $\varphi_2 = 45°\mathrm{N}$, $\varphi_3 = 55°\mathrm{N}$, $\tau_1 = 0.0748\,\mathrm{N\,m}^{-2}$, and $\tau_2 = 0.0249\,\mathrm{N\,m}^{-2}$. This wind stress yields the Sverdrup streamfunction depicted in Fig. 6a, in which the Sverdrup transport zero is given along 45°N. Miyama et al.[24] discussed numerical results of other idealized cases.

**Meteorological Research Institute Community Ocean Model.** Figure 7 shows the barotropic streamfunction simulated by a western North Pacific version of the Meteorological Research Institute Community Ocean Model (MRI.COM)[33], which is a free-surface, depth-coordinate ocean general circulation model, spanning zonally in 117°E–160°W and meridionally in 15°N–65°N. Both zonal and meridional resolution is 1/10° in the domain 117°E–160°E, 15°N–50°N, and 1/6° in 160°E–160°W, 50°N–65°N. It has 54 levels with the interval increasing from 1 m at the surface to 250 m near the bottom. The bottom topography is based on Smith and Sandwell[52]. The model was driven by the momentum, heat, and fresh water flux estimated from the atmospheric reanalysis dataset produced by the National Center for Environmental Prediction and the National Center for Atmospheric Research (NCEP-R1)[53] in the simulation depicted in Fig. 7[34]. The streamfunction is the average during the period 1987–2006.

**Eddy tracking.** It was suggested that eddy propagation that is deflected from due west is attributed to ambient barotropic flows[35]. This motivated us to evaluate quantitatively barotropic flow from the eddy propagation features.

The propagation speeds of mesoscale eddies in Fig. 8 were calculated by tracking SSH anomalies provided by AVISO using a software tool (py-eddy-tracker) that enables the identification and automated tracking of oceanic eddies[54]. The input SSH anomalies were the two-sat-merged, delayed-time global reference gridded daily data from AVISO (AVISO 2013) during 1993–2012. Eddies with lifetimes greater than 28 days were stored. The propagation speeds of the eddies were averaged into 0.5° × 0.5° horizontal grids. A 5-point weighted smoothing (4 times at the central grid) was applied to the gridded velocities.

**Data availability.** All data are available from the authors on reasonable request.

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

## Acknowledgements

This work was benefitted by fruitful discussions with Drs. B. Taguchi and Y. Yoshikawa. The western North Pacific version of MRI.COM is kindly provided by Dr. H. Tsujino. This work was supported by the Ministry of Education, Culture, Sports, Science and Technology, Japan, Grants-in-Aid for Scientific Research (22221001, 26247076, JP16H01585, 17H00775), and by the Joint Research Program of the Institute of Low Temperature Science, Hokkaido University.

## Author contributions

H.M. designed and conceived the study, and wrote the manuscript with input from all coauthors. T.M., T.N. and H.N. analyzed and interpreted the data, and H.N. helped with theory. T.M. also conducted the numerical experiments. Y.F. provided with a numerical model output. T.N., T.W., R.F. and S.I. discussed and interpreted the results. All authors wrote the manuscript.

## Additional information

**Competing interests:** The authors declare no competing interests.

