## [Peer Review File(PDF 540 kb) · Nature Communications]

Reviewers' comments:

Reviewer #1 (Remarks to the Author):

In the western North Pacific, the Subarctic Frontal Zone at the confluence of the subarctic and subtropical gyres is characterised by sharp horizontal gradients of sea-surface temperature (SST). This region is of great interest because of its regional and even long-range influence on climate variability. However, there is a need for better understanding of the oceanic processes driving and maintaining these sharp temperature gradients.

This article focusses on the potential role of recently identified quasi-stationary jets in the Subarctic Frontal Zone. These jets (in particular "Isoguchi jet J1") flow northwestward and transport warm water from the Kuroshio Extension into contact with cold subarctic water from the Oyashio Current. The first part of the paper makes a convincing case for the relationship between the SST gradients in the region and the strength of the J1 jet, as measured by the change in SSH across the jet.

The second contribution of the article is to propose a possible formation mechanism for these jets. In particular, the authors attempt to resolve a puzzling feature, namely, the formation of jets aligned in a non-zonal direction away from a western boundary. A comparison with bottom topography does suggest that jet-topographic interactions may play a role. However, the topographic features here are relatively small (~500m) compared to the depth of the Pacific in this region (~6000m).

The solution proposed by the authors is an intriguing one. By examining Rossby wave characteristics, the authors suggest that the jet is formed by a thickness jump generated by long wave baroclinic Rossby waves that are steered by barotropic flow. The barotropic flow forms because of the presence of closed contours of f/H . In this way, relatively small (low) topographic features can impact jet and front formation in subpolar regions.

This is a plausible suggestion and one that could have fascinating implications for the role of low topographic features in otherwise flat regions of the subpolar oceans, in particular the Pacific sector of the ACC. However, I feel that the authors have not yet sealed the deal. Although the mechanism is plausible, the two-layer model used to elucidate the mechanism is over-simplified. The parameters of the model are reasonable, but there is no sensitivity analysis presented. What of the role of eddies? And can this mechanism explain the formation of the second jet, J2?

In conclusion, I recommend major revision. This is an intriguing idea but I would like to see more confirmation of the hypothesis. One possibility would be to repeat the analysis with the topographic features removed. Do the jets form in that case? Another possibility would be idealized numerical model studies, although that might be asking too much of a short communication.

Referee's Report for *Nature Comm.* on **Low ocean-floor rises regulate subpolar sea surface temperature by forming baroclinic jets** by *Mitsudera et al*

July 19, 2017

This paper aims to explain the presence of quasi-stationary jets/fronts in the Kuroshio extension which seem somewhat anchored to bottom topography inspite of the fact that the bottom topography in question is relatively small in vertical extent compared to the the ocean depth. The principle dynamic pathway suggest by the authors is as follows:

1. Presence of closed f/H contours in the region collocated with the presence of Kuroshio eddy forcing which through a process of PV homogenization like dynamics forms a barotropic flow in the region.
2. The barotropic flow significantly modifies the *background* pv gradient compared to the usual terms and modifies the propagation of long-wave baroclinic Rossby waves(RW), and correspondingly, mesoscale eddies which generally propagate at long wave baroclinic Rossby speeds.
3. The hyperbolic points of Q_c determine locations of strong eddy/wave flux convergence and a strong jet forms collocated with this convergence.

I am generally quite happy with both the approach and the detail of the content of this paper and am willing to recommend publication. But I have important questions and changes that I want addressed or reasons given as to why the authors should retain the present structure.

1. While discussing point 2 above, the authors start directly with the baroclinic RW propagation, with no context or motivation as to why this is done. The point about mesoscale eddies almost having RW speeds must be added here alongwith the relevant

references. This point is an area of active recent research over the past 7 years and cannot be treated as obvious even though theoretical underpinnings go as far as the 1980s.

2. I want a longer discussion about the role of hyperbolic points and eddy flux convergence either in the main body or in the Supplementary information. This is currently insufficiently unmotivated (inspite of references provided) and has a bit of a feel of a mystery novel, which while intriguing fails somewhat in paper format. In general even though the authors do their eddy tracking and show that it seems to follow the characteristic lines, a critical point, this should have been the motivation all along. This should not be mentioned as an afterthought, but as the *primary reason*. (more on this below)
3. Along the lines of the previous point, a normal statistic that one likes to see in this case is the E-P flux divergence with relative contributions from various terms that comprise it. The authors' hypothesis that this happens because of eddy-thickness flux convergence at **h1** should be supported by this metric in the Supplementary info. I personally think that this discussion is more important than the various lagged covariances of SST and SSH indices.
4. Based on Figure 1, and as explained in the text, two branches of the jet are at J1 and J2 with their respective topographic associations. Variability increases and mean velocity decreases in the region between. The structure of J1 is the focus of this paper. But what about J2? Since I don't see a hyperbolic point in the vicinity of J2, what generates that jet? I don't expect the authors to solve every problem in one paper but since time has been spent explaining J1 and introducing J2 by them, I would like some speculation in the paper as to how and if the dynamics at J2 is different, or at least a note to that effect. This is another reason to look at the EP flux divergence.
5. Speaking of eddy paths, Figure 3 d is not clear and I don't see the point being made. There are a lot of arrows in regions not representing the characteric curves and they make things unclear. We need to see a clear association between eddy tracks and characteristic curves. That is not evident and must be clarified.
6. What is the relative strength of the background barotropic velocity and the surface intensified jet speed? A figure to this effect in the supplementary info might be of value.

Review of **Low ocean-floor rises regulate subpolar sea surface temperature by forming baroclinic jets**

In the article *Low ocean-floor rises regulate subpolar sea surface temperature by forming baroclinic jets* by H. Mitsudera et al, the authors investigate the mechanisms driving the formation of quasi-stationary jets and related SST fronts in the western subarctic Pacific. The SST fronts formed in this region have a strong impact on the atmosphere, but their generation mechanisms are not well understood yet.

Here the authors use an ocean reanalysis product to show the correlation between the SST fronts and the presence of quasi-stationary jets. The formation of the jets is explained using a theoretical model of the dynamics of Rossby waves that propagate along deformed characteristics curves due to the local barotropic circulation and bring subpolar and subtropical water together. The barotropic circulation is formed by anticyclonic circulations around low rises forming closed geostrophic contours.

This represents a new mechanism that highlights the impact of the deep ocean topography on the ocean circulation and the atmosphere. The article suggests that the same mechanism may explain the localizations of jets and SST fronts by deep topography elsewhere, and in particular in the Southern Ocean where they are numerous. The paper is well written and details significant new work, therefore I recommend publication after the following concerns are addressed.

The authors do not make a difference between eddies and Rossby waves. They assume that propagation speeds of mesoscale eddies are the same than baroclinic Rossby waves. However this is not always true (e.g. Chelton et al. (2011)) and should be discussed here.

The barotropic currents from the reanalysis are scaled by a factor of 0.4 to account for the difference with observed eddy propagation speed. This is a significant difference that should require further investigation. Have the authors looked at the barotropic currents in this region in other models, reanalysis products or observations (e.g. Colin de Verdière and Ollitrault (2016)) to see if it matches exactly the eddy propagation speed?

Minor comments:

- 1.2, remove *It has become evident that*
- 1.6, *...can form in the ocean interior away from the western boundary*. The ocean interior often refers to the ocean beneath the surface mixed-layer. I suggest using another expression or leaving only *...can form away from the western boundary*. (see also for 1.27, 1.71, 1.131)
- 1.13 *formidable* seems a little excessive. Please rephrase.
- 1. 37 it is puzzling
- 1.119 *although a clockwise circulation in the subarctic gyre is somewhat counter-intuitive*. I don't understand what the authors are referring to here. Are they just stating that one would expect cyclonic circulation north of the jet and anticyclonic circulation

south of the jet? If this is the case, rephrase.

- 1.121 In the SAFZ
- 1.180: a better understanding
- 1.197: are derived
- 1.198: was calculated
- 1.378: analyses

REFERENCES

- Chelton, D. B., M. G. Schlax, and R. M. Samelson, 2011: Global observations of non-linear mesoscale eddies. *Progress in Oceanography*, **91** (2), 167 – 216, , URL <http://www.sciencedirect.com/science/article/pii/S00796661111000036>.
- Colin de Verdière, A. and M. Ollitrault, 2016: A direct determination of the world ocean barotropic circulation. *J. Phys. Oceanogr.*, **46**, 255273.

Reply to reviewers' comments

Reviewer#1:

We would like to deeply appreciate the reviewer's thoughtful and constructive comments. We believe that the manuscript has been much improved. Major changes responding to the comments and questions that are common to all reviewers' concern are as follows.

Major changes

1. The structure of the paper was reorganized. After discussing SST and the J1 relationship, we included a new section (L.79-116) concerning the propagation of decadal SSH signals and their convergence at the J1. Associated with this change, the new Fig. 3 was included. Then, we proceeded to the discussion on the baroclinic Rossby wave characteristics and their convergence, followed by the discussion on the roles of barotropic flows. We believe that the motivation of this paper becomes clearer by this reorganization.
2. Numerical experiments on the sensitivity to the presence or absence of a low topographic rise (L.211-227 and Fig. 6) were included. A manuscript of a related work (Miyama et al., submitted to Journal of Physical Oceanography) is attached for reference. Although we utilized the model setting shown in Fig. 12b of Miyama et al., the analyses and the sensitivity experiment on the presence/absence of topography, shown by Fig. 6 of the present paper, is new. Further, to bolster the discussion on the role of barotropic flows over topography, another numerical-modeling example was included (Fig. 7).
3. The unperturbed baroclinic Rossby wave phase speed was reconsidered by referring to the Rossby radius of deformation by Killworth et al. (1997). Associated with this reconsideration, we removed the scaling factor 0.4 when evaluating the barotropic flow speed.
4. Characteristic curves of baroclinic Rossby wave were reconsidered in this revision for steep topographic features; a brief derivation is included in Methods. Detailed derivation and discussions are presented in Miyama et al. (attached manuscript). Nevertheless, since we have dealt with gentle small-amplitude topography, discussion and conclusion are not changed by this reconsideration of Rossby wave characteristics.

Point-by-point responses to the reviewer's comments follow:

This is a plausible suggestion and one that could have fascinating implications for the role of low topographic features in otherwise flat regions of the subpolar oceans, in particular the Pacific sector of the ACC. However, I feel that the authors have not yet sealed the deal. Although the

mechanism is plausible, the two-layer model used to elucidate the mechanism is over-simplified. The parameters of the model are reasonable, but there is no sensitivity analysis presented. What of the role of eddies? And can this mechanism explain the formation of the second jet, J2?

In conclusion, I recommend major revision. This is an intriguing idea but I would like to see more confirmation of the hypothesis. One possibility would be to repeat the analysis with the topographic features removed. Do the jets form in that case? Another possibility would be idealized numerical model studies, although that might be asking too much of a short communication.

We would appreciate the reviewer's constructive comments and questions. According to the reviewer's comments, we described a result of simple numerical experiments in L.211-227 and Fig. 6 using a two-layer version of HYCOM. This numerical experiment indicates that in the presence of a rise with a height of 500 m, a surface jet corresponding to the J1 forms on the eastern flank of the rise of the rise, while it disappears in the absence of the rise. The mechanism of the jet formation in the numerical experiment is explained well by the characteristic curves in the same way as the J1 formation in the ocean reanalysis JCOPE2.

The analyses and the sensitivity experiment shown in Fig. 6 of the present paper is new, although we utilized the model setting shown in Fig. 12b of Miyama et al. We would note that the meridional basin scale of the present model (Fig. 6) is suited for the North Pacific Ocean compared with that of Miyama et al., because most of experiments in Miyama et al. were done in a narrow basin with a small meridional extent to discuss purely idealized situations. In the present model, the subtropical western boundary current is separated from the western boundary at 35 °N as seen in our Fig. 6b, which mimics the Kuroshio Extension well. Therefore, the relative locations between the western boundary current and the seafloor rise in the present model is close to the relationship between the Kuroshio Extension and the real seafloor rise, that is, the location of the non-zonal oceanic jet in reality. This simulation supports our hypothesis. Therefore, we believe that this simulation would satisfy the reviewer's request. We also included another numerical simulation in Fig. 7 to confirm the role of barotropic flows over the small-amplitude rise A.

Further, we would like to point out that the decadal satellite SSH signals (lag correlations) in Fig. 3 follow closely to the characteristic curves derived from equation (2). As described in a new section (L.79-116), Figure 3 shows the convergence of the satellite-derived SSH anomaly of the subarctic origin and that of subtropical origin at the J1. This also supports the validity of our hypothesis.

The formation of the J2 is briefly discussed in the conclusion section (L.267-278). The

mechanism of the J2 formation is similar to that of the J1, in which the J2 is related to the rise C (Fig. 1b) and the hyperbolic point h2 (Fig. 4a). The J2 formation is caused by the layer thickness jump across the characteristic curve going out from h2 in Fig. 4a.

We believe that the above evidences sufficiently prove the mechanism of the non-zonal jet formation away from the western boundary through baroclinic Rossby wave characteristics as proposed in this paper.

Reviewer #2

We would appreciate the reviewer's thoughtful and constructive comments. We believe that the manuscript has been much improved. Major changes responding to the comments and questions that are common to all reviewers' concern are as follows.

Major changes

1. The structure of the paper was reorganized. After discussing SST and the J1 relationship, we included a new section (L.79-116) concerning the propagation of decadal SSH signals and their convergence at the J1. Associated with this change, the new Fig. 3 was included. Then, we proceeded to the discussion on the baroclinic Rossby wave characteristics and their convergence, followed by the discussion on the roles of barotropic flows. We believe that the motivation of this paper becomes clearer by this reorganization.
2. Numerical experiments on the sensitivity to the presence or absence of a low topographic rise (L.211-227 and Fig. 6) were included. A manuscript of a related work (Miyama et al., submitted to Journal of Physical Oceanography) is attached for reference. Although we utilized the model setting shown in Fig. 12b of Miyama et al., the analyses and the sensitivity experiment on the presence/absence of topography, shown by Fig. 6 of the present paper, is new. Further, to bolster the discussion on the role of barotropic flows over topography, another numerical-modeling example was included (Fig. 7).
3. The unperturbed baroclinic Rossby wave phase speed was reconsidered by referring to the Rossby radius of deformation by Killworth et al. (1997). Associated with this reconsideration, we removed the scaling factor 0.4 when evaluating the barotropic flow speed.
4. Characteristic curves of baroclinic Rossby wave were reconsidered in this revision for steep topographic features; a brief derivation is included in Methods. Detailed derivation and discussions are presented in Miyama et al. (attached manuscript). Nevertheless, since we have dealt with gentle small-amplitude topography, discussion and conclusion are not changed by this reconsideration of Rossby wave characteristics.

Point-by-point responses to the reviewer's comments follow.

This paper aims to explain the presence of quasi-stationary jets/fronts in the Kuroshio extension which seem somewhat anchored to bottom topography inspite of the fact that the bottom topography in question is relatively small in vertical extent compared to the ocean depth. The principle dynamic pathway suggest by the authors is as follows:

1. Presence of closed f/H contours in the region collocated with the presence of Kuroshio eddy

forcing which through a process of PV homogenization like dynamics forms a barotropic flow in the region.

2. The barotropic flow significantly modifies the background pv gradient compared to the usual terms and modifies the propagation of long-wave baroclinic Rossby waves (RW), and correspondingly, mesoscale eddies which generally propagate at long wave baroclinic Rossby speeds.

3. The hyperbolic points of Q_c determine locations of strong eddy/wave flux convergence and a strong jet forms collocated with this convergence.

I am generally quite happy with both the approach and the detail of the content of this paper and am willing to recommend publication. But I have important questions and changes that I want addressed or reasons given as to why the authors should retain the present structure.

We would thank the reviewer for the constructive comments. We reorganized the structure of this paper as stated in the “Major changes” above, in accord with the reviewer’s comments.

1. While discussing point 2 above, the authors start directly with the baroclinic RW propagation, with no context or motivation as to why this is done. The point about mesoscale eddies almost having RW speeds must be added here along with the relevant references. This point is an area of active recent research over the past 7 years and cannot be treated as obvious even though theoretical underpinnings go as far as the 1980s.

We included a new section on the propagation of decadal SSH anomalies (L.79-116) before discussing the baroclinic Rossby wave characteristics. A lag correlation analysis of the satellite-derived SSH anomaly (Fig. 3) clearly indicates the convergence of decadal SSH signals at the J1. We have found the decadal SSH signals and characteristic curves are related closely with each other. We did not use the result of eddy tracking here because the decadal satellite SSH is a direct association with the baroclinic Rossby wave propagation as pointed out.

2. I want a longer discussion about the role of hyperbolic points and eddy flux convergence either in the main body or in the Supplementary information. This is currently insufficiently unmotivated (in spite of references provided) and has a bit of a feel of a mystery novel, which while intriguing fails somewhat in paper format. In general even though the authors do their eddy tracking and show that it seems to follow the characteristic lines, a critical point, this should have been the motivation all along. This should not be mentioned as an afterthought, but as the primary reason. (more on this below)

According to this reviewer's comment, we reorganized the structure of this paper. We believe that the paper is now improved substantially by including the description on the convergence of the decadal satellite SSH signals in the new section (L.79-116) and Fig. 3 as mentioned above. We believe that this provides a sufficient motivation (L.111-116) to use the Rossby wave characteristics to solve this problem. We also added a discussion on the relationship between the convergence of the decadal SSH propagation and the hyperbolic point h1 (L.171-177) .

3. Along the lines of the previous point, a normal statistic that one likes to see in this case is the E-P flux divergence with relative contributions from various terms that comprise it. The authors' hypothesis that this happens because of eddy-thickness flux convergence at h1 should be supported by this metric in the Supplementary info. I personally think that this discussion is more important than the various lagged covariances of SST and SSH indices.

Blocking of the westward propagating Rossby waves at h1 were discussed by the satellite-derived decadal SSH anomaly (L.171-177). As mentioned above, and as pointed out by the reviewer, the analysis of the decadal SSH anomaly is more relevant than the eddy-tracking analysis from a point of view of the Rossby wave characteristics. Further, eddy potential vorticity fluxes were calculated using the formula of equation (A10) in Methods and displayed in Fig. 4d. Eddy thickness flux dominates over other terms of equation (A10) by a factor of 10^2 . Since the eddy flux vectors and their convergence/divergence field are small scale features, we decided to focus on describing them in the vicinity of the J1 (Fig.4d and L.185-195). As Fig. 4d indicates, the eddy thickness flux tends to intensify the J1 downstream of h1 because it is divergent (the upper layer thinning) northward of the jet and convergent (the upper layer thickening) southward.

With respect to the eddy-tracking analysis, it seems that the previous manuscript might confuse the reviewer. Our intention of eddy tracking was not for the sake of showing the eddy thickness flux convergence at h1 but showing the presence of barotropic flows by observing the meridional deflection of the eddy movement. In this revision, we clarified this point by moving the description of the eddy-tracking analysis to the barotropic flow section (L.236-240) and including Fig. 8.

4. Based on Figure 1, and as explained in the text, two branches of the jet are at J1 and J2 with their respective topographic associations. Variability increases and mean velocity decreases in the region between. The structure of J1 is the focus of this paper. But what about J2? Since I don't see a hyperbolic point in the vicinity of J2, what generates that jet? I don't expect the authors to solve every problem in one paper but since time has been spent explaining J1 and introducing J2 by

them, I would like some speculation in the paper as to how and if the dynamics at J2 is different, or at least a note to that effect. This is another reason to look at the EP flux divergence.

The formation of the J2 is discussed in the conclusion section (L.267-278). The mechanism of the J2 formation is similar to that of the J1, in which the J2 is related to the rise C (Fig. 1b) and the hyperbolic point h2 (Fig. 4a). The J2 is caused by the layer thickness jump across the characteristic curve going out from h2 in Fig. 4a. An important aspect of the J2 formation is the southwestward deflection of the relatively-thin layer on the northern side of this characteristic curve from h2; such a southwestward propagation is captured by the decadal SSH anomaly as in Fig. 3a (e.g. lag=-2yr). Therefore, the relatively thin layer intrudes into the region where the thick layer of the subtropical origin is present, thereby causing a thickness jump across the characteristic curve originating at h2. This leads to the formation of the J2, which is similar to the J1 formation.

5. Speaking of eddy paths, Figure 3 d is not clear and I don't see the point being made. There are a lot of arrows in regions not representing the characteristic curves and they make things unclear. We need to see a clear association between eddy tracks and characteristic curves. That is not evident and must be clarified.

As mentioned above, the propagation of decadal SSH signals exhibits a direct association with the characteristic curves as shown by Fig. 3a and b. A new section describing the decadal SSH is included in L.79-116, and relationship between the decadal SSH propagation and characteristic curves are discussed in L.168-174, and L.267-271.

6. What is the relative strength of the background barotropic velocity and the surface intensified jet speed? A figure to this effect in the supplementary info might be of value.

Figure S4 is included in the supplementary information. We can observe intensification of the J1 and the J2 on the eastern flank of the rise A and the rise C, respectively.

Reviewer#3

We would like to deeply appreciate the reviewer's thoughtful and constructive comments. We believe that the manuscript has been much improved. Major changes responding to the comments and questions that are common to all reviewers' concern are as follows.

Major changes

1. The structure of the paper was reorganized. After discussing SST and the J1 relationship, we included a new section (L.79-116) concerning the propagation of decadal SSH signals and their convergence at the J1. Associated with this change, the new Fig. 3 was included. Then, we proceeded to the discussion on the baroclinic Rossby wave characteristics and their convergence, followed by the discussion on the roles of barotropic flows. We believe that the motivation of this paper becomes clearer by this reorganization.
2. Numerical experiments on the sensitivity to the presence or absence of a low topographic rise (L.211-227 and Fig. 6) were included. A manuscript of a related work (Miyama et al., submitted to Journal of Physical Oceanography) is attached for reference. Although we utilized the model setting shown in Fig. 12b of Miyama et al., the analyses and the sensitivity experiment on the presence/absence of topography, shown by Fig. 6 of the present paper, is new. Further, to bolster the discussion on the role of barotropic flows over topography, another numerical-modeling example was included (Fig. 7).
3. The unperturbed baroclinic Rossby wave phase speed was reconsidered by referring to the Rossby radius of deformation by Killworth et al. (1997). Associated with this reconsideration, we removed the scaling factor 0.4 when evaluating the barotropic flow speed.
4. Characteristic curves of baroclinic Rossby wave were reconsidered in this revision for steep topographic features; a brief derivation is included in Methods. Detailed derivation and discussions are presented in Miyama et al. (attached manuscript). Nevertheless, since we have dealt with gentle small-amplitude topography, discussion and conclusion are not changed by this reconsideration of Rossby wave characteristics.

Point-by-point responses to the reviewer's comments follow.

The authors do not make a difference between eddies and Rossby waves. They assume that propagation speeds of mesoscale eddies are the same than baroclinic Rossby waves. However this is not always true (e.g. Chelton et al. (2011)) and should be discussed here.

In this revised manuscript, we presented the analysis of the satellite-derived, decadal SSH signals,

which is more relevant than the eddy tracking analysis from a point of view of the baroclinic Rossby wave characteristics. We have found a close association between the characteristic curves and the propagation of decadal SSH signals. To describe this, we included a new section (L.79-116) and Fig. 3. The result of eddy tracking analysis was not used for this purpose.

Our intention of the eddy tracking analysis was to show the presence of barotropic flows over the rise A-A' in reality by showing the meridional deflection of the eddy movement. In this revision, we made this point clearer by discussing the meridional deflection referring to Chelton et al. (2011). Correspondingly, the description of the eddy tracking analysis was moved to the barotropic flow section (L.236-250 and Fig. 8).

The barotropic currents from the reanalysis are scaled by a factor of 0.4 to account for the difference with observed eddy propagation speed. This is a significant difference that should require further investigation. Have the authors looked at the barotropic currents in this region in other models, reanalysis products or observations (e.g. Colin de Verdiere and Ollitrault (2016)) to see if it matches exactly the eddy propagation speed?

In the revised manuscript, the scaling factor 0.4 to the model barotropic flow was removed. We reconsidered the unperturbed baroclinic Rossby wave propagation based on the Rossby radius given in Killworth et al. (1997). With this revision, the characteristic curves show clear association with the propagation of decadal SSH signals (Fig. 3). As noted above, the eddy tracking was conducted for the sake of showing the meridional deflection of the eddy propagation, which is likely caused by the ambient barotropic flow as discussed by Chelton et al. (2011). Further, another numerical-modeling example of the barotropic flow field is shown in Fig. 7. We believe that the clockwise barotropic flow over the rise A is a robust feature.

Minor comments:

_ 1.2, remove *It has become evident that*

Removed.

_ 1.6, *...can form in the ocean interior away from the western boundary*. The ocean interior often refers to the ocean beneath the surface mixed-layer. I suggest using another expression or leaving only *...can form away from the western boundary*. (see also for 1.27,1.71, 1.131)

“in the ocean interior” is removed.

_ 1.13 *formidable* seems a little excessive. Please rephrase.

L.13 Changed to “overlooked role”

_ 1. 37 it is puzzling

L.37 Changed as suggested.

_ 1.119 *although a clockwise circulation in the subarctic gyre is somewhat counter-intuitive.* I dont understand what the authors are referring to here. Are they just stating that one would expect cyclonic circulation north of the jet and anticyclonic circulation south of the jet? If this is the case, rephrase.

This phrase was removed.

_ 1.121 In the SAFZ

L.247 Changed as suggested.

_ 1.180: a better understanding

The sentence was removed.

_ 1.197: are derived

L.309 Corrected.

_ 1.198: was calculated

L.309 Corrected

_ 1.378: analyses

L.530 Corrected.

REVIEWERS' COMMENTS:

Reviewer #1 (Remarks to the Author):

This article has greatly improved with the addition of, in particular, the simple schematic presented in Fig 4c and the idealized numerical simulations in Fig 6, both of which aid the reader in understanding the underlying mechanism.

As I wrote in my original review, the implications of this mechanism for ACC in the Pacific sector are potentially quite excited. I encourage the authors to pursue this line of enquiry. I will be happy to review their next Nature article on this topic!

Below are some minor typos I spotted. I don't need to see this article again until it is in print.

Shane Keating, UNSW Sydney, Australia

L88: "even though these boxes are ONLY 100 KM APART from each other."

L153: "although equation (2) does not include THE topographic term explicitly..."

L224 and L427: "Miyama et al., attached manuscript". Is Miyama et al. intended to be added to the supplementary material? If not, it should be referred to as "Miyama et al., in preparation" or whatever is appropriate for Nature Communications.

L292: "the rise IS likely to have significant influences..."

L421: "A NON-SLIP BOUNDARY CONDITION is applied..."

Reviewer #2 (Remarks to the Author):

I think the present manuscript and the the additions of the discussion of the Eddy-thickness flux in terms of the expanded equations in the appendices A4-A9 and the addition of the idealized two-layer model does help clarify the central claims of the Jet formation at J1-2. Figure 4(d) that would show an eddy-thickness flux convergence isn't actually as clear as I would have liked it to be or as much as the authors imply it is, and the choice of a red marker in a a blue-red colormap is a poor one. But as it stands, I think this manuscript has sufficient scientific content and narrative structure to be published. So I accept.

Review of **Low ocean-floor rises regulate subpolar sea surface temperature by forming baroclinic jets**

I am happy with the changes that were made to the manuscript. The discussion of SSH signals and the addition of new numerical experiments significantly strengthen the conclusions of the paper.

I point out a few typos and corrections below. More generally there are parts of the manuscript where the writing could be improved (especially the parts that were recently added). So I suggest a careful checking of the manuscript.

I recommend publication of the manuscript once these (very minor) issues are addressed.

- 1.21, the SAFZ
- 1.53, we note that
- 1.83. the North Pacific
- 1.90 and fig.3: space between number and unit, year instead of yr
- 1.103, SSH north of
- 1.108, remove thus
- 1.114, baroclinic
- 1.120, motivate
- 1.144, remove: because of the joint effect of the baroclinicity and bottom relief (JEBAR). Generally, JEBAR is not a cause (except if you prescribe it as a forcing in a barotropic model). In this case, this term is just a result of the baroclinic structure of the flow.
- 1.212. elevation of 500 m
- 1.214, remove is located
- 1.218, of subarctic origin
- 1.219, curves (yellow contour in Fig.6b))
- 1.225, the topography
- 1.230, is high enough

2

- 1.232, such as the community ..., where the isoguchi J1 is also reproduced.
- 1.237, remove: although it is important for the J1 formation
- 1.248, eddies on the western flank of the rise B propagate northward... consistently with the characteristic curves.
- 1.255, western boundary, and why it is collocated with small amplitude rises...
- 1.267, we have focused
- 1.276, the same as the formation mechanism of the J1
- 1.292, likely has
- 1.298, remove: in a manner similar to the J1 formation

Reply to the reviewers' comments

We would like to thank reviewers for their constructive comments. We have revised the manuscript considering all of the comments. We would also appreciate pointing out typos.

Reviewer #1 (Remarks to the Author):

Below are some minor typos I spotted. I don't need to see this article again until it is in print.

L88: "even though these boxes are ONLY 100 KM APART from each other."

L.132: revised as indicated

L153: "although equation (2) does not include THE topographic term explicitly..."

L.199: added "the".

L224 and L427: "Miyama et al., attached manuscript". Is Miyama et al. intended to be added to the supplementary material? If not, it should be referred to as "Miyama et al., in preparation" or whatever is appropriate for Nature Communications.

The paper is now in press in J. Phys. Oceanogr. It is referred to as "24".

L292: "the rise IS likely to have significant influences..."

L.342: added "is".

L421: "A NON-SLIP BOUNDARY CONDITION is applied..."

L.473: Changed as suggested.

Reviewer #2 (Remarks to the Author):

I think the present manuscript and the the additions of the discussion of the Eddy-thickness flux in terms of the expanded equations in the appendices A4-A9 and the addition of the idealized two-layer model does help clarify the central claims of the Jet formation at J1-2. Figure 4(d) that would show an eddy-thickness flux convergence isn't actually as clear as I would have liked it to be or as much as the authors imply it is, and the choice of a red marker in a a blue-red colormap is a poor one. But as it stands, I think this manuscript has sufficient scientific content and narrative structure to be published. So I accept.

I would appreciate the reviewer's comment. The marker in Fig. 4d is changed from red

to yellow.

Reviewer #3 (Remarks to the Author):

I recommend publication of the manuscript once these (very minor) issues are addressed.

_ 1.21, the SAFZ

L.51: added “the”

_ 1.53, we note that

L.95: added “that”

_ 1.83. the North Pacific

L.126: added “the”

_ 1.90 and _g.3: space between number and unit, year instead of yr

L.133 and Fig. 3: changed as indicated.

_ 1.103, SSH north of

L.146: changed as indicated.

_ 1.108, remove thus

L.152: removed.

_ 1.114, baroclinic

L.158: corrected.

_ 1.120, motivate

L.164: corrected

_ 1.144, remove: because of the joint effect of the baroclinicity and bottom relief (JEBAR). Generally, JEBAR is not a cause (except if you prescribe it as a forcing in a barotropic model). In this case, this term is just a result of the baroclinic structure of the flow.

L.190: removed as indicated.

_ 1.212. elevation of 500 m

L.258: corrected.

_ 1.214, remove is located

L.260: removed.

_ 1.218, of subarctic origin

L.264: removed “the”

_ 1.219, curves (yellow contour in Fig.6b))

L.265: changed as suggested.

_ 1.225, the topography

L.270: added “the”

_ 1.230, is high enough

L.276: corrected.

_ 1.232, such as the community ..., where the isoguchi J1 is also reproduced.

L.277: Changed as suggested.

_ 1.237, remove: although it is important for the J1 formation

L.283: removed as suggested.

_ 1.248, eddies on the western flank of the rise B propagate northward... consistently with the characteristic curves.

L.293: changed as indicated.

_ 1.255, western boundary, and why it is collocated with small amplitude rises...

L.302: changed as indicated.

_ 1.267, we have focused

L.316: added “have”

_ 1.276, the same as the formation mechanism of the J1

L.325: changed as indicated.

_ 1.292, likely has

L.342: changed to “is likely to have”

_ 1.298, remove: in a manner similar to the J1 formation

L.348: removed as indicated.